# Diffusion-Weighted Imaging Can Differentiate between Malignant and Benign Pleural Diseases

**DOI:** 10.3390/cancers11060811

**Published:** 2019-06-12

**Authors:** Katsuo Usuda, Shun Iwai, Aika Funasaki, Atsushi Sekimura, Nozomu Motono, Munetaka Matoba, Mariko Doai, Sohsuke Yamada, Yoshimichi Ueda, Hidetaka Uramoto

**Affiliations:** 1Department of Thoracic Surgery, Kanazawa Medical University, Ishikawa 920-0293, Japan; mhg1214@kanazawa-med.ac.jp (S.I.); aicarby@kanazawa-med.ac.jp (A.F.); a24seki@kanazawa-med.ac.jp (A.S.); motono@kanazawa-med.ac.jp (N.M.); hidetaka@kanazawa-med.ac.jp (H.U.); 2Department of Radiology, Kanazawa Medical University, Ishikawa 920-0293, Japan; m-matoba@kanazawa-med.ac.jp (M.M.); doaimari@kanazawa-med.ac.jp (M.D.); 3Department of Pathology and Laboratory Medicine, Kanazawa Medical University, Ishikawa 920-0293, Japan; sohsuke@kanazawa-med.ac.jp; 4Department of Pathophysiological and Experimental Pathology, Kanazawa Medical University, Ishikawa 920-0293, Japan; z-ueda@kanazawa-med.ac.jp

**Keywords:** magnetic resonance imaging, diffusion-weighted imaging, malignant pleural mesothelioma, pleural dissemination, empyema, pleural effusion

## Abstract

It is not clear whether magnetic resonance imaging (MRI) is useful for the assessment of pleural diseases. The aim of this study is to determine whether diffusion-weighted magnetic resonance imaging (DWI) can differentiate malignant pleural mesothelioma (MPM) from pleural dissemination of lung cancer, empyema or pleural effusion. The DWI was calibrated with the b value of 0 and 800 s/mm^2^. There were 11 MPMs (8 epithelioid and 3 biphasic), 10 pleural disseminations of lung cancer, 10 empyemas, and 12 pleural effusions. The apparent diffusion coefficient (ADC) of the pleural diseases was 1.22 ± 0.25 × 10^−3^ mm^2^/s in the MPMs, 1.31 ± 0.49 × 10^−3^ mm^2^/s in the pleural disseminations, 2.01 ± 0.45 × 10^−3^ mm^2^/s in the empyemas and 3.76 ± 0.62 × 10^−3^ mm^2^/s in the pleural effusions. The ADC of the MPMs and the pleural disseminations were significantly lower than the ADC of the empyemas and the pleural effusions. Concerning the diffusion pattern of DWI, all 11 MPMs showed strong continuous diffusion, 9 of 10 pleural disseminations showed strong scattered diffusion and 1 pleural dissemination showed strong continuous diffusion, all 10 empyemas showed weak continuous diffusion, and all 12 pleural effusions showed no decreased diffusion. DWI can evaluate pleural diseases morphologically and qualitatively, and thus differentiate between malignant and benign pleural diseases.

## 1. Introduction

Malignant pleural mesothelioma (MPM) is a highly lethal disease with limited therapeutic options that are difficult for patients to endure. Computed tomography (CT) is the primary imaging modality used to evaluate MPM and accurately track the spread of the primary tumor, the intrathoracic lymphadenopathy, as well as the extrathoracic spread. Early diagnosis of MPM is usually difficult in the modality of a chest CT, because pleural thickness or pleural effusion due to MPM can resemble a benign disease on a CT scan image. Additional imaging modalities, such as magnetic resonance imaging (MRI) and fluoro-2-deoxy-glucose positron emission tomography/computed tomography (FDG-PET/CT), have emerged in recent years and are complementary to CT for disease staging and evaluation of patients with MPM [1]. There have been several recent research papers that have shown that MRI is the best modality for the assessment of the pleural interface, characterization of complex pleural effusion, and identification of exudate and hemorrhages, and staging accuracy [2]. MRI is more accurate than CT in staging evaluation of local invasions in the endothoracic fascia or single chest wall focus (68% vs. 46%) and the diaphragm (82% vs. 55%) [3]. Diffusion-weighted magnetic resonance imaging (DWI) has reportedly been useful in differentiating malignant pleural diseases from benign alterations [4]. DWI utilizes the random, translational motion, or so-called Brownian movement, of water molecules in biologic tissue [5]. DWI has been used for brain imaging, mainly for the assessment of acute ischemic strokes, intracranial tumors and demyelinating diseases [6].

The purpose of this study was to determine whether DWI can differentiate MPM from pleural dissemination of lung cancer, empyema, or pleural effusion.

## 2. Results

Below is shown typical radiological imaging of an MPM (Figure 1), a pleural dissemination of lung cancer (Figure 2), an empyema (Figure 3) and a pleural effusion (Figure 4). 

Diffusion patterns based on pleural lesions are shown in Table 1. All 11 cases of MPM showed strong continuous diffusion. Nine of the 10 pleural disseminations showed strong scattered diffusion, and one pleural dissemination showed strong continuous diffusion. All 10 empyemas showed weak continuous diffusion. All 12 pleural effusions showed no decreased diffusion. 

The ADC values of the pleural diseases were 1.22 ± 0.25 × 10^−3^ mm^2^/s in the MPMs, 1.31 ± 0.49 × 10^−3^ mm^2^/s in the pleural disseminations, 2.01 ± 0.45 × 10^−3^ mm^2^/s in the empyemas and 3.76 ± 0.62 × 10^−3^ mm^2^/s in the pleural effusions (Figure 5). The ADC of the MPMs was significantly lower than that of the empyemas (P = 0.0007) and the pleural effusions (P < 0.0001). The ADC values of the pleural disseminations were significantly lower than that of the empyemas (P = 0.0086) and the pleural effusions (P < 0.0001). The ADC of the MPM was not significantly different from that of pleural dissemination.

There were eight epithelioid and three biphasic MPMs. The ADC (1.23 ± 0.26 × 10^−3^ mm^2^/s) of the epithelioid MPMs was not significantly higher than the ADC (1.17 ± 0.21 × 10^−3^ mm^2^/s) of the biphasic MPMs (P = 0.73).

Of the 10 pleural disseminations of lung cancer, seven were adenocarcinomas, two were small cell carcinomas, and one was large cell neuroendocrine carcinoma (LCNEC). The ADC of the pleural dissemination was 1.63 ± 0.13 × 10^−3^ mm^2^/s in adenocarcinomas, 0.68 ± 0.18× 10^−3^ mm^2^/s in small cell carcinomas, and 0.67 × 10^−3^ mm^2^/s in a LCNEC. The ADC of adenocarcinomas was significantly higher than that of small cell carcinomas (P = 0.0002).

Six of the pleural effusions were exudative, two were caused by atelectasis, two were caused by malignant uterus tumors, one was caused by trauma, and one was caused by asbestosis. Because the two patients with malignant uterus tumors had negative results of pleural cytology and did not have pleural dissemination, the causes of pleural effusion in the malignant uterus tumors were not clear. 

The ADC values of the pleural fluid were 3.87 ± 0.29 × 10^−3^ mm^2^/s in MPMs, 3.59 ± 0.57 × 10^−3^ mm^2^/s in the pleural disseminations, 2.40 ± 1.26 × 10^−3^ mm^2^/s in the empyemas, and 3.94 ± 0.27 × 10^−3^ mm^2^/s in the pleural effusions (Figure 6). The ADC of pleural fluid was significantly lower in the empyemas than in the MPMs (P = 0.0023), the pleural disseminations (P = 0.0286) and the pleural effusions (P = 0.0009). No significant differences in pleural fluid were found in MPMs, pleural disseminations, and pleural effusions.

## 3. Discussion

DWI can be used to detect the restricted diffusion of water molecules in the body. Evidence has shown the advantage of DWI for functional information of neoplasm [7]. It has been reported that DWI has the potential for differential diagnosis of pulmonary nodules and masses, and evaluation of N factor/M factor/stage of lung cancer [8,9,10,11]. Two reports of meta-analysis concluded that DWI was an accurate modality to evaluate N factor of lung cancer [12,13].

Diagnosing MPM correctly in its early phases is very difficult with CT because of fewer symptoms, less pleural thickness or pleural effusion. At this stage, a chest CT would be performed to assess the pleural disease, but the MPM might not be detected. MRI is an important adjunctive imaging examination in thoracic oncologic imaging that is applied as a problem-solving tool to evaluate chest wall invasion, intraspinal extension, and cardiac/vascular invasion [14]. Moreover, functional MR imaging modality, such as DWI and dynamic contrast material–enhanced MR imaging modality, have already proven effective in differentiating malignant from benign pleural diseases and assessing chest wall and diaphragmatic involvement [15,16,17,18,19]. As this research has shown, DWI can diagnose an MPM as a malignant tumor with strong continuous diffusion over the entire circumference of the pleural with an ADC of less than 1.70 × 10^−3^ mm^2^/s. The MPMs present marked restricted diffusion of the pleural tumor in DWI [20].

Coolen et al. [4] reported that visual evaluation of pleural pointillism was denoted by the presence of multiple, hyperintense pleural spots on high-b-value DWI. It is useful to differentiate MPM from benign alterations, which perform substantially better than mediastinal pleural thickness and shrinking lung. Pleural pointillism could be used to provide guidance for biopsies and thoracoscopic evaluations. In our article, one MPM also had hyperintense pleural diffusion on DWI, which we expressed as “strong continuous diffusion“. Strong continuous diffusion and pleural pointillism can be seen in DWI of MPMs. Although pleural pointillism on high-*b*-value DWI had better sensitivity and accuracy than pleural thickness and lung shrinkage, the specificity was not significantly better, which suggests that the three parameters should complement, rather than substitute, one another. Gill et al. [19] also reported that the ADC value (1.31 ± 0.15 × 10^−3^ mm^2^/s) of the epithelioid MPMs was statistically higher than that (1.01 ± 0.11 × 10^−3^ mm^2^/s) of the biphasic MPMs (P = 0.00024). In our study, we could not repeat the results from Gill’s article due to small sample size. 

MRI has advantages over FDG-PET/CT due to its excellent soft tissue contrast and absence of ionizing radiation [7]. Uptake of FDG in FDG-PET/CT is not always tumor specific, as seen in benign inflammatory lesions, which can make accurate staging of MPMs difficult [21]. Although FDG-PET/CT is more effective than CT for differentiating malignant from benign pleural diseases [22], its utility is limited to assessing the primary tumor extension and the nodal status of MPM [23]. In general, FDG-PET/CT can detect thicker parts of MPMs, but not thinner parts of MPMs.

DWI could diagnose pleural dissemination from lung cancer as a malignant tumor with strong scattered diffusion along the pleura when the ADC was less than 1.70 × 10^−3^ mm^2^/s. Pleural metastasis from breast cancer can be detected with DWI [24]. Moreover, DWI can evaluate pleural lesions morphologically and qualitatively, while also differentiating between malignant pleural diseases and benign pleural diseases. We have also shown that DWI can differentiate MPM from pleural dissemination.

In our cases of empyema, The ADC of the pleural effusion decreased due to pus. The ADC of pleural fluid in empyemas was significantly lower than the ADC of the MPMs, the pleural disseminations, and the pleural effusions. Abscesses and thrombi impede the diffusivity of water molecules due to their hyperviscous nature [7,25]. Therefore, the ADC value of empyema was significantly lower without a significant decreased diffusion of the pleura. To diagnose empyema, the ADC value of pleura and pleural cavity must be carefully evaluated.

MRI has proven superior to CT in the characterization of pleural fluid [26]. Pleural fluid is typically hypointense on T1-weighted imaging and hyperintense on T2-weighted imaging [2]. As shown in this article, when a patient with pleural fluid shows no decreased diffusion in DWI, and when the ADC of the pleural fluid is higher, the pleural disease can be identified as benign. 

MRIs have several advantages. They do not involve contrast agents and examinations take less time. Furthermore, MRIs do not have the downside of radiation exposure and are suitable for children with negative reactions to radiation [27]. On the other hand, an MRI is not a viable option for people with devices, such as pacemakers, or tattoos. In addition, MRI examinations are loud, and can cause some patients anxiety.

### Limitations

There are two important limitations in this study. First, this study was performed at a single institution and dealt with a small number of patients, which unavoidably introduced selection bias. Second, we could not acquire consent forms from all the patients who developed pleural lesions in the period from March 2015 to February 2019. Due to these limitations, our sample size was small. Our ADC measurements were repeated three times and the minimum ADC value was recorded. Thus, this ADC may not fully represent the whole tumor. There is no overwhelming agreement in the literature concerning optimal DWI techniques and image analysis procedures, including region of interest (ROI) size and placement.

Nevertheless, DWI has two limitations. First, some of the more common benign diseases which present decreased diffusion and lower ADC values in DWI are pulmonary abscesses and empyemas with histopathological necrosis. Second, mucinous adenocarcinomas were usually hypointense in DWI and showed higher ADC values, which could be misjudged as benign lesions in DWI [28]. Since mucinous adenocarcinomas had lower cellularity than tubular adenocarcinomas, it had lower DWI signal intensity and higher ADC values than tubular adenocarcinoma in the ano-rectal region [29].

## 4. Patients and Methods

### 4.1. Eligibility

The study protocol for evaluating pleural lesions with CT and MRI with DWI was approved by the ethical committee of Kanazawa Medical University (the approval number: No.189, the approval date: 25th January 2012). After discussing the risks and benefits of the examinations with their surgeons, we obtained written informed consents for CT and MRI from each patient.

### 4.2. Patients

A total of 43 patients with pleural diseases or pleural effusions underwent CT scans and MRI examinations and were enrolled in this study in the period from March 2015 to February 2019. Exclusion criteria were unwillingness to undergo an MRI examination, and the general issues that exclude most patients from getting an MRI (e.g., incompatible implanted medical devices, claustrophobia, and tattoos).

The diagnosis of MPM and pleural dissemination of lung cancers were confirmed histopathologically by biopsy or surgical procedure. The diagnosis of empyema was confirmed by positive culture of pleural effusion or by purulent pleural effusion. The diagnosis of pleural effusion was confirmed by negative cytology and negative culture of pleural effusion. 

There were 11 MPMs, 10 pleural disseminations of lung cancers, 10 empyemas and 12 pleural effusions. 

### 4.3. Magnetic Resonance Imaging (MRI)

The 1.5 T superconducting magnetic scanner (Magnetom Avanto; Siemens, Erlangen, Germany) with two anterior six-channel body phased-array coils and two posterior spinal clusters (six-channels each) was used for MRI examinations. The conventional MRI consisted of a coronal T1-weighted spin-echo sequence and coronal and axial T2-weighted fast spin-echo sequences. DWI using a single-shot echo-planar technique were performed under SPAIR (spectral attenuated inversion recovery) with respiratory triggered scan with the following parameter: b value = 0 and 800 s/mm^2^; TR/TE/flip angle, 3000–4500/65/90; diffusion gradient encoding in three orthogonal directions; receiver bandwidth, 2442 Hz/Px; voxel size, 2.7 × 2.7 × 6.0 mm; field of view, 350 mm; matrix size, 128 × 128; number of excitations, 5; section thickness, 6 mm and section gap, 0 mm. After image reconstruction, a 2-dimensional (2D) round or elliptical ROI was marked on the lesion which was detected visually on the ADC map with reference to T1-weighted or CT image by a radiologist (M.D.) with 26 years of MRI experience who was unaware of the patients’ clinical data. The procedure was done three times and we recorded the minimum apparent diffusion coefficient (ADC) value. A consensus was reached if there were any differences of opinion. The optical cutoff value (OCV) of ADC for diagnosing malignancy in DWI was determined to be 1.70 × 10^−3^ mm^2^/s using the receiver operating characteristics (ROC) curve, as previously described [8]. Pleural lesions with an ADC value of the same or less than the OCV were classified as positive. Pleural lesions with ADC value of more than the OCV or those that could not be detected on DWI were classified as negative. Diffusion pattern of DWI were divided into 4 categories: ① strong continuous diffusion, ② strong scattered diffusion, ③ weak continuous diffusion and ④ no decreased diffusion. 

FDG-PET/CTs were also used for the evaluation. FDG-PET/CTs were done with a dedicated PET camera (SIEMENS Biograph Sensation 16, Erlangen Germany). All patients fasted for 6 hours before the scans. The OCV of maximum standardized uptake value (SUVmax) for diagnosing malignancy in FDG-PET/CT was determined to be 4.45 using the ROC curve, as previously described [8].

### 4.4. Statistical Analysis

Using StatView for Windows (Version 5.0; SAS Institute Inc. Cary, NC, USA), statistical analysis was performed. The data is presented as the mean ± standard deviation. A two-tailed Student t test was used for comparison of ADC values of various pleural lesions. A p-value of <0.05 was deemed statistically significant. 

## 5. Conclusions

DWI can evaluate pleural diseases morphologically and qualitatively, while also differentiating between malignant pleural diseases and benign pleural diseases. DWI has the ability to differentiate MPM from pleural dissemination.

## Figures and Tables

**Figure 1 cancers-11-00811-f001:**
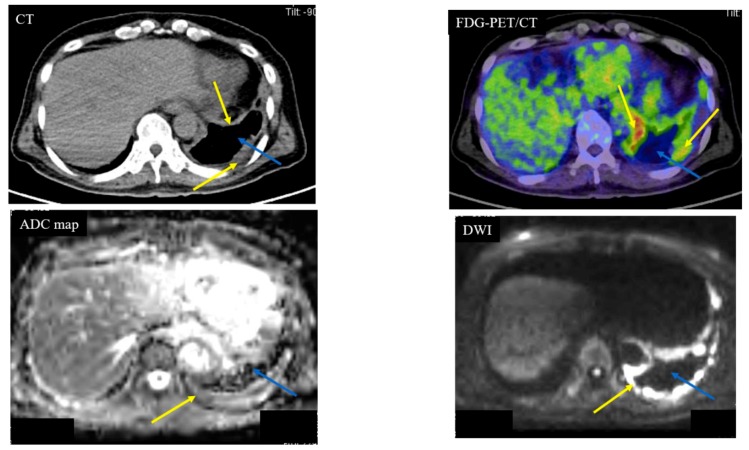
Case 1—Malignant Pleural Mesothelioma (MPM). A 64-year-old male with MPM (cT4N2M0). The yellow arrows indicate the MPM. The blue arrows indicate pleural effusion. Computed tomography (CT) showed left pleural thickness of the MPM. The apparent diffusion coefficient (ADC) of the MPM was 0.84 × 10^−3^ mm^2^/s (positive) and the ADC of the pleural fluid was 3.95 × 10^−3^ mm^2^/s (negative). Fluorodeoxyglucose (FDG)-position emission tomography (PET)/CT showed partial accumulation (standardized uptake value (SUV)max: 12.39) of FDG on the MPM.

**Figure 2 cancers-11-00811-f002:**
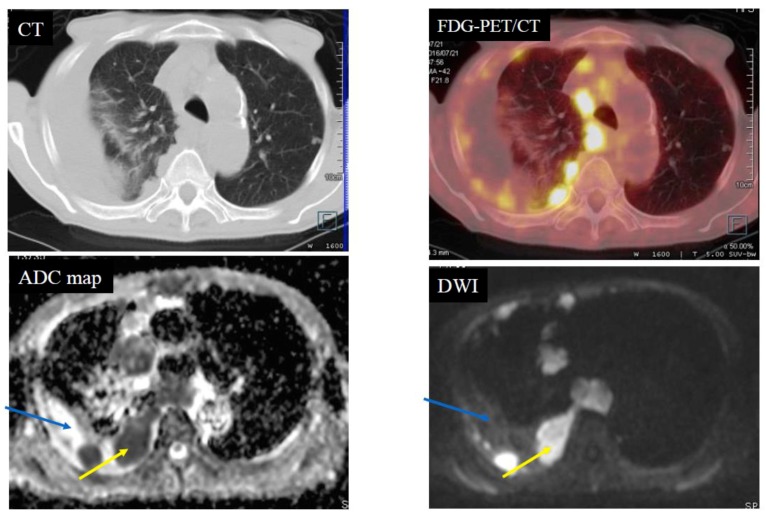
Case 2—Pleural dissemination of lung cancer. An 81-year-old male with pleural dissemination of a large cell neuroendocrine carcinoma. The yellow arrows indicate pleural dissemination. The blue arrows indicate pleural fluid. The apparent diffusion coefficient (ADC) of the pleural dissemination was 0.67 × 10^−3^ mm^2^/s (positive) and the ADC of the pleural fluid was 3.03 × 10^−3^ mm^2^/s (negative). Fluorodeoxyglucose-position emission tomography/computed tomography (FDG-PET/CT) showed scattered accumulation (standardized uptake value (SUVmax): 14.7) of the FDG on the pleural dissemination.

**Figure 3 cancers-11-00811-f003:**
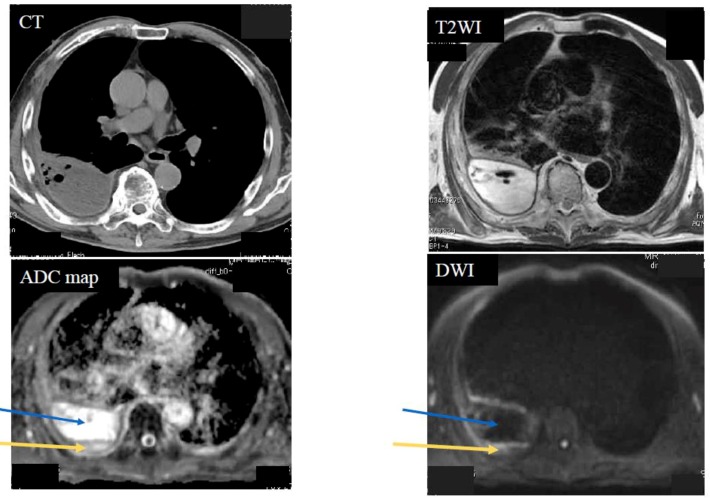
Case 3—Empyema. A 70-year-old male with right empyema. The yellow arrows indicate pleural thickness. The blue arrows indicate pleural fluid. The apparent diffusion coefficient (ADC) of the pleural thickness was 1.82 × 10^−3^ mm^2^/s (negative) and the ADC of the pleural fluid was 3.95 × 10^−3^ mm^2^/s (negative).

**Figure 4 cancers-11-00811-f004:**
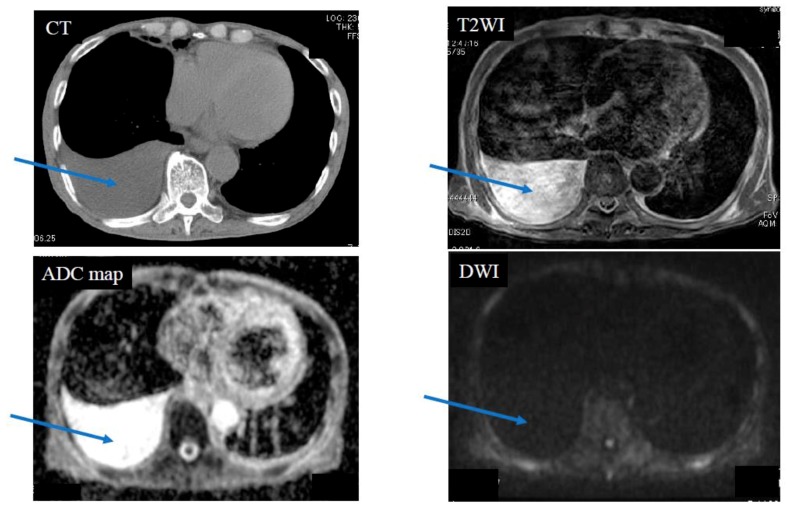
Case 4—Pleural effusion due to exudative pleurisy in a 79-year-old male, who suffered from right pneumonia. The blue arrows indicate pleural fluid. Pleural effusion was not seen in diffusion weighted magnetic resonance imaging (DWI). The apparent diffusion coefficient (ADC) of pleural fluid was 4.02 × 10^−3^ mm^2^/s (negative).

**Figure 5 cancers-11-00811-f005:**
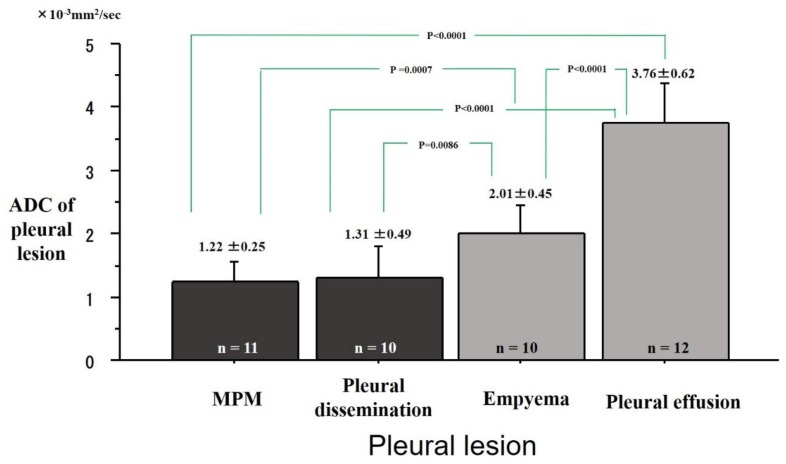
Differences of apparent diffusion coefficient (ADC) of pleural lesions. Mean ADCs were 1.22 ± 0.25 × 10^−3^ mm^2^/s in Malignant Pleural Mesothelioma (MPM), 1.31 ± 0.49 × 10^−3^ mm^2^/s in pleural dissemination, 2.01 ± 0.45 ×10^−3^ mm^2^/s in empyema, and 3.76 ± 0.62 × 10^−3^ mm^2^/s on pleural effusion. The ADC of the MPM was significantly lower than that of empyema (P = 0.0007) or pleural effusion (P < 0.0001). The ADC of pleural dissemination was significantly lower than that of empyema (P = 0.0086) or pleural effusion (P < 0.0001).

**Figure 6 cancers-11-00811-f006:**
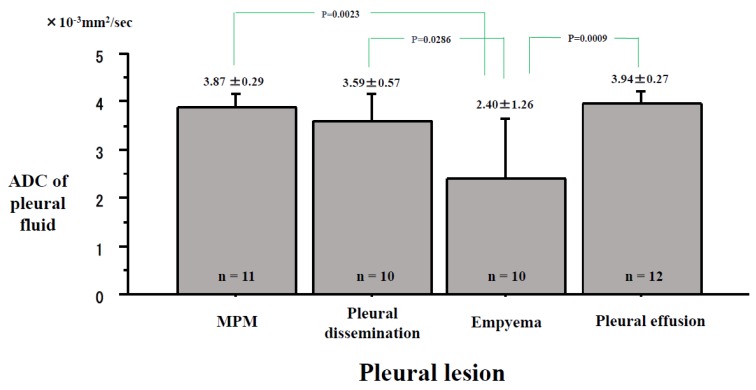
Differences in the apparent diffusion coefficient (ADC) of the pleural effusion inside the pleural lesions. The ADCs of pleural fluid were 3.87 ± 0.29 × 10^−3^ mm^2^/s in Malignant Pleural Mesothelioma (MPM), 3.59 ± 0.57 × 10^−3^ mm^2^/s in the pleural dissemination, 2.40 ± 1.26 × 10^−3^ mm^2^/s in the empyema, and 3.94 ± 0.27 × 10^−3^ mm^2^/s in the pleural effusion.

**Table 1 cancers-11-00811-t001:** Diffusion patterns of DWI for pleural lesions.

Diffusion Pattern		Strong Continuous	Strong Scattered	Weak Continuous	No Decreased	No. of Cases
Diagnosis	MPM	11	0	0	0	11
	Pleural dissemination	1	9	0	0	10
	Empyema	0	0	10	0	10
	Pleural effusion	0	0	0	12	12
No. of cases		12	9	10	12	43

MPM: Malignant Pleural Mesothelioma.

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
