# Peer review of "Diffusion-Weighted Imaging Can Differentiate between Malignant and Benign Pleural Diseases"

_cancers, 2019, doi:10.3390/cancers11060811_

Round 1

Reviewer 1 Report

The authors have provided convincing evidence to demonstrate the capability of MRI to differentiate between Malignant and benign Pleural Diseases. However, could the authors also specify whether each case belongs to benign or Malignant for each reported case? 

Author Response

Comments and Suggestions for Authors

The authors have provided convincing evidence to demonstrate the capability of MRI to differentiate between Malignant and benign Pleural Diseases. However, could the authors also specify whether each case belongs to benign or malignant for each reported case?   

 The data that we presented clearly shows whether each case is malignant or benign.

Reviewer 2 Report

The manuscript presents an interesting topic, considering the uncertainties regarding the clinical utility of MRI in the diagnostics of pleural diseases and the frequent difficulties encountered by clinicians, radiologists and pathologists in diagnosing these diseases. In particular, diffusion-weighted magnetic resonance imaging (DWI) could potentially allow earlier diagnosis of malignant pleural mesothelioma (MPM) and thereby could contribute to prompter therapeutic intervention and improved outcomes for this very aggressive cancer type. However, the manuscript requires adjustments before being suitable for publication.

SPECIFIC COMMENTS

The title of the manuscript seems a bit redundant and can be shortened a little. For instance, “Diffusion-Weighted Imaging Can Differentiate between Malignant and Benign Pleural Diseases” might be more suitable.

Introduction, line 39-41, “Early diagnosis of MPM ….. , because pleural thickness or pleural effusion in CT looks a benign disease” should be rephrased more accurately. For ex., “Early diagnosis of MPM …… , because pleural thickness or pleural effusion due to MPM may resemble a benign disease in CT”.

The materials used in the study should be explained in more detail. The authors state that they have investigated 11 MPMs. However, it would be appropriate to indicate the histological subtypes of the analyzed MPMs and whether there were differences in DWI and ADC from subtype to subtype. Abstract, Materials and Methods, Results and possibly Discussion, should be adjusted accordingly. In this respect, comparisons with the results presented in reference 19 should be discussed.

They also write that “10 pleural disseminations” were used. This should be better explained. In the Abstract, it should be specified that they were disseminations from lung cancer. Moreover, it would be proper to indicate in the Materials and Methods and Results, which histological types of lung cancer were analyzed (in Figure 2 an ex. of pleural spreading from a LCNEC is presented), as their biology and structure differ and may influence the results of the MR. Thus, possible differences among lung cancer types, if any, should be described.

The 12 “effusions” the authors have investigated should be clarified as well. As also shown in this manuscript, both malignant and benign pleural diseases can produce effusions. The authors should define better what the term indicates in their paper. The impression is that they mean benign/reactive effusions due to inflammatory conditions or others, but this needs to be specified more clearly throughout the manuscript (including Abstract).

Line 181-182 of Patients and methods: The sentence “The patients included in this study were selected according to clinically indicated CT” is a bit ambiguous. It should be reformulated more clearly.

Line 184-185, “Diagnosis of MPM and pleural dissemination of lung cancers were confirmed cytopathologically”: The final diagnosis of MPM is histological. By examining biopsies and surgical samples the diagnosis was histopathological not cytopathological. In addition, how were effusions diagnosed/confirmed?

In figure legends to figure 1-4 the word "presents" should be replaced by the more appropriate word "indicates" or similar.

Figure 1: the blue arrows do not seem to be oriented correctly (towards the chest wall instead of the effusion). In addition, CT alone should also be described in the legend.

Figure 4: Specify in the legend what was the cause of the pleural effusion.

Results, line 84-85, “The ADC of MPM was significantly lower than that of empyema (P = 0.0007) or pleural effusion (P < 0.0001)”: Please indicate also that the ADC of MPMs and pleural disseminations were not significantly different.

Similarly, line 96-98, “The ADC of pleural fluid in empyemas was significantly lower than that in MPMs (P = 0.0023), that in pleural disseminations (P = 0.0286) and than that in pleural effusions (P = 0.0009)”: it would be appropriate to add that no significant differences were found among pleural fluid in MPMs, pleural disseminations, and effusions.

References: Reference 4 and 21 are the same. The citations in the text and list of references should be adjusted accordingly. In addition, small typos in reference 3, 4, 14, 18, 21, and 25 should be corrected.

Discussion: Since the journal Cancers has a broad readership, it would be appropriate to briefly define the concept of pleural pointillism for the readers unfamiliar with the term.

Line 119-123, “MPM presents …. invasive procedures for MPM [21]”: In ref. 21 (i.e., 4), the authors showed that pointillism had higher sensitivity and accuracy than pleural thickness and lung shrinkage, while the specificity was not significantly different. This suggests that the three parameters should complement, rather than substitute, one another.

Line 131-132, “In general, FDG-PET/CT can detect some parts of MPM, but cannot 131 detect the area where few MPM exist”. The sentence should be rephrased more clearly. Do the authors mean that PET/CT cannot detect small MPMs or else?

The two sentences on line 133 to 136 are redundant, they repeat each other.

Line 148, “shows no decrease diffusion”: should it be “shows no decreased diffusion”?

MINOR POINTS

Abstract, line 22: The abbreviation ADC should be defined the first time. Right now, one needs to read the methods on page 8 to find the abbreviation spelled out.

Patients and methods, line 204: The abbreviation OCV should be defined/spelled out the first time.

Line 219: correct “resions” to “lesions”

Author Response

Comments and Suggestions for Authors

The manuscript presents an interesting topic, considering the uncertainties regarding the clinical utility of MRI in the diagnostics of pleural diseases and the frequent difficulties encountered by clinicians, radiologists and pathologists in diagnosing these diseases. In particular, diffusion-weighted magnetic resonance imaging (DWI) could potentially allow earlier diagnosis of malignant pleural mesothelioma (MPM) and thereby could contribute to prompter therapeutic intervention and improved outcomes for this very aggressive cancer type. However, the manuscript requires adjustments before being suitable for publication.

SPECIFIC COMMENTS

The title of the manuscript seems a bit redundant and can be shortened a little. For instance, “Diffusion-Weighted Imaging Can Differentiate between Malignant and Benign Pleural Diseases” might be more suitable.

  We changed the title according to your advice: Diffusion-Weighted Imaging Can Differentiate between Malignant and Benign Pleural Diseases

Introduction, line 39-41, “Early diagnosis of MPM ….. , because pleural thickness or pleural effusion in CT looks a benign disease” should be rephrased more accurately. For ex., “Early diagnosis of MPM …… , because pleural thickness or pleural effusion due to MPM may resemble a benign disease in CT”.

 We changed the sentence according to your advice:  Early diagnosis of MPM is usually difficult in the recent modality of chest CT, because pleural thickness or pleural effusion due to MPM may resemble a benign disease in CT.

The materials used in the study should be explained in more detail. The authors state that they have investigated 11 MPMs. However, it would be appropriate to indicate the histological subtypes of the analyzed MPMs and whether there were differences in DWI and ADC from subtype to subtype. Abstract, Materials and Methods, Results and possibly Discussion, should be adjusted accordingly. In this respect, comparisons with the results presented in reference 19

 should be discussed.

   There were 8 epithelioid and 3 biphasic MPMs. The ADC (1.23±0.26 × 10-3mm2/sec) of the epithelioid MPMs was not significantly lower than that (1.38±0.47 × 10-3mm2/sec) of the biphasic MPMs (P =0.51). Although Gill RR et al. [19] reported that the ADC value (1.31±0.15 × 10-3mm2/sec) of the epithelioid MPMs was statistically higher than that (1.01±0.11 × 10-3mm2/sec) of the biphasic MPMs (P=0.00024), we recognize there were some outliers in our data but due to the small sample size no conclusive results could be determined.    

They also write that “10 pleural disseminations” were used. This should be better explained. In the Abstract, it should be specified that they were disseminations from lung cancer. Moreover, it would be proper to indicate in the Materials and Methods and Results, which histological types of lung cancer were analyzed (in Figure 2 an ex. of pleural spreading from a LCNEC is presented), as their biology and structure differ and may influence the results of the MR. Thus, possible differences among lung cancer types, if any, should be described.

There were 7 adenocarcinomas, 2 small cell carcinomas, and 1 large cell neuroendocrine carcinoma. The ADC value of pleural disseminations of lung cancer were 1.63±0.13× 10-3mm2/sec in adenocarcinomas, 0.68±0.18× 10-3mm2/sec in small cell carcinomas, and 0.67 × 10-3mm2/sec in a LCNEC. The ADC of adenocarcinomas was significantly higher than that of small cell carcinomas (P =0.0002).

The 12 “effusions” the authors have investigated should be clarified as well. As also shown in this manuscript, both malignant and benign pleural diseases can produce effusions. The authors should define better what the term indicates in their paper. The impression is that they mean benign/reactive effusions due to inflammatory conditions or others, but this needs to be specified more clearly throughout the manuscript (including Abstract). 

Causes of the pleural effusion were exudative pleurisy in 6 patients, pleural effusion due to atelectasis in 2, pleural effusion related to uterus malignant tumors in 2, pleural effusion due to trauma in one, and pleural effusion due to benign asbestos in one. 

Line 181-182 of Patients and methods: The sentence “The patients included in this study were selected according to clinically indicated CT” is a bit ambiguous. It should be reformulated more clearly.

A total of 43 consecutive patients with pleural disease or pleural effusion underwent CT scans and MRI examination and were enrolled in this study in the period from March 2015 to February 2019.  

Line 184-185, “Diagnosis of MPM and pleural dissemination of lung cancers were confirmed cytopathologically”: The final diagnosis of MPM is histological. By examining biopsies and surgical samples the diagnosis was histopathological not cytopathological. In addition, how were effusions diagnosed/confirmed?

Yes, the diagnosis of MPM and pleural dissemination of lung cancers were confirmed histopathological by biopsy or surgical procedure. Pleural effusion was confirmed when cytology of the pleural effusion was negative and its culture was negative.  We unintentionally misused the terminology and appreciate you pointing that out.

In figure legends to figure 1-4 the word "presents" should be replaced by the more appropriate word "indicates" or similar.

According the reviewer’s advice, I replaced the word "presents" to " indicates ".

Figure 1: the blue arrows do not seem to be oriented correctly (towards the chest wall instead of the effusion). In addition, CT alone should also be described in the legend.

That was our formatting mistake.  According to the reviewer’s advice, I changed the direction of the blue arrows.

Figure 4: Specify in the legend what was the cause of the pleural effusion.

Pleural effusion due to exudative pleurisy.

Results, line 84-85, “The ADC of MPM was significantly lower than that of empyema (P = 0.0007) or pleural effusion (P < 0.0001)”:  Please indicate also that the ADC of MPMs and pleural disseminations were not significantly different.

The ADC of MPM was not significantly different from that of pleural dissemination.

Similarly, line 96-98, “The ADC of pleural fluid in empyemas was significantly lower than that in MPMs (P = 0.0023), that in pleural disseminations (P = 0.0286) and than that in pleural effusions (P = 0.0009)”: it would be appropriate to add that no significant differences were found among pleural fluid in MPMs, pleural disseminations, and effusions.

As to the reviewer’s comment, I added that no significant differences were found among pleural fluid in MPMs, pleural disseminations, and effusions.

References: Reference 4 and 21 are the same. The citations in the text and list of references should be adjusted accordingly. In addition, small typos in reference 3, 4, 14, 18, 21, and 25 should be corrected.

The reviewers advice was correct.  I deleted Reference 21, and fixed the small typos in reference 3, 4, 14, 18, 21, and 25.

Discussion: Since the journal Cancers has a broad readership, it would be appropriate to briefly define the concept of pleural pointillism for the readers unfamiliar with the term.

The reviewer offered very practical advice.  For readers unfamiliar with the term pleural pointillism we have better defined it in the paper

Coolen J. et al [4] reported that visual evaluation of plural pointillism was denoted by the presence of multiple, hyperintense pleural spots on high-b-value DWI. It is useful to differentiate MPM from benign alterations, performing substantially better than mediastinal pleural thickness and shrinking lung. Plural pointillism might be used to provide guidance for biopsy or thoracoscopic evaluation. On our article, an MPM also had hyperintense pleural diffusion on DWI, and expressed as “Strong continuous diffusion.“  Strong continuous diffusion as well as plural pointillism can be seen in DWI of MPMs.

Line 119-123, “MPM presents …. invasive procedures for MPM [21]”: In ref. 21 (i.e., 4), the authors showed that pointillism had higher sensitivity and accuracy than pleural thickness and lung shrinkage, while the specificity was not significantly different. This suggests that the three parameters should complement, rather than substitute, one another.

I agree with the reviewer’s opinion.  Although pleural pointillism on high-b-value DWI had a better sensitivity and accuracy than pleural thickness and lung shrinkage, the specificity was not significantly better than either of them: which suggests that the three parameters should complement, rather than substitute, one another.

Line 131-132, “In general, FDG-PET/CT can detect some parts of MPM, but cannot detect the area where few MPM exist”. The sentence should be rephrased more clearly. Do the authors mean that PET/CT cannot detect small MPMs or else?

As to the reviewer’s comment, FDG-PET/CT can detect thicker parts of MPMs, but not thinner parts of MPMs.

The two sentences on line 133 to 136 are redundant, they repeat each other.

Line 148, “shows no decrease diffusion”: should it be “shows no decreased diffusion”?

As to the reviewer’s comment, the sentence was deleted: DWI showed disseminated pleural tumors as strong scattered diffusion along to the pleura. ADC of disseminated pleural tumors was less than 1.70×10-3mm2/sec.

I changed it to “shows no decreased diffusion”.

MINOR POINTS

Abstract, line 22: The abbreviation ADC should be defined the first time. Right now, one needs to read the methods on page 8 to find the abbreviation spelled out.

As to the reviewer’s comment, apparent diffusion coefficient (ADC) is added in the Patients and Methods as well as in the abstract.

s

Patients and methods, line 204: The abbreviation OCV should be defined/spelled out the first time.

  optical cutoff value (OCV) was added in Patients and methods.

Line 219: correct “resions” to “lesions”

  I corrected “resions” to “lesions”.

Round 2

Reviewer 2 Report

The authors have properly addressed most of the previous comments. There are the following minor points that remain to be amended before publication.

Abstract, line 20, “10 pleural disseminations”: Specify that they were from lung cancer, as in theory many cancer types can disseminate to the pleural (directly or via metastasis).

Figure legend 1: For completeness, it should probably be specified that it is an MPM of the left pleura and CT alone should also be briefly described.

Figure legend 3, line 71: change “Blue arrow presents” to “Blue arrow shows”.

Figure legend 4, line 74-5: it should be “Pleural effusion due to exudative pleurisy in a 79-year-old male, who suffered from right pneumonia”.

Results, line 93-95: shorten the sentence by eliminating the repetition "pleural effusion" (repeated 5 times in the sentence …). "Causes of pleural effusion were exudative ... , atelectasis, malignant uterus tumor, trauma ..., asbestosis …”. With respect to the meaning of the sentence, it is a bit unclear whether “related to uterus malignant tumor” is due to pleural metastases from uterine cancer or to infection/inflammation caused by uterine cancer. Perhaps the authors could specify for clarity. Moreover, on line 95 “benign asbestos” seems to be a bit confusing a definition (what is benign asbestos??). The authors should use asbestosis instead, which is known as a benign disease, as opposed to MPM.

Discussion, Line 132, 135, and 137: correct “plural” to “pleural”. Line 140: change to "of them, which suggests...".

Line 144-5, “we recognize there were some outliers in our data but due to the small sample size no 144 conclusive results could be determined”: This somehow contradicts the Results (line 87-89) stating that ADC of epithelioid and biphasic MPM were not significantly different. Rephrase the sentence less contradictorily.

Methods, line 205: confirmed histopathologically.

Line 205-6, “Pleural effusion was confirmed when cytology of the pleural effusion was negative and its culture was negative”: rephrase the sentence more clearly, right now it does not make much sense. For what was the effusion negative for? And for what was its culture negative for?

Author Response

Dustin Keeling, who is an American and owns an English language service company, has proofread this document and the journal being submitted. 

Abstract, line 20, “10 pleural disseminations”: Specify that they were from lung cancer, as in theory many cancer types can disseminate to the pleural (directly or via metastasis).

 The 10 pleural disseminations were from lung cancer.

Figure legend 1: For completeness, it should probably be specified that it is an MPM of the left pleura and CT alone should also be briefly described.

 As to the reviewer’s advice, I added yellow arrows which indicated MPM on CT.  CT showed left pleural thickness of MPM.

Figure legend 3, line 71: change “Blue arrow presents” to “Blue arrow shows”.

 I changed “Blue arrow presents” to “Blue arrows indicate” according to other figure legends.

Figure legend 4, line 74-5: it should be “Pleural effusion due to exudative pleurisy in a 79-year-old male, who suffered from right pneumonia”.

 As to the reviewer’s advice, I changed the title of the figure legend to “ Pleural effusion due to exudative pleurisy in a 79-year-old male, who suffered from right pneumonia”.

Results, line 93-95: shorten the sentence by eliminating the repetition "pleural effusion" (repeated 5 times in the sentence …). "Causes of pleural effusion were exudative ... , atelectasis, malignant uterus tumor, trauma ..., asbestosis …”. With respect to the meaning of the sentence, it is a bit unclear whether “related to uterus malignant tumor” is due to pleural metastases from uterine cancer or to infection/inflammation caused by uterine cancer. Perhaps the authors could specify for clarity. Moreover, on line 95 “benign asbestos” seems to be a bit confusing a definition (what is benign asbestos??). The authors should use asbestosis instead, which is known as a benign disease, as opposed to MPM.

⇒ Causes of pleural effusion were exudative in 6, atelectasis in 2, malignant uterus tumor in 2, trauma in 1 and asbestosis in 1 patient.  The two patients of a malignant uterus tumor had negative results of pleural cytology and did not have pleural dissemination, the causes of pleural effusion in the malignant uterus tumors were not clear.   As to the reviewer’s advice, I used the word “asbestosis.”

Discussion, Line 132, 135, and 137: corrected “plural” to “pleural”.

 I corrected “plural” to “pleural” on Line 132, 135, and 137.    

Line 140: changed to "of them, which suggests...".

 The specificity was not significantly better than either of sensitivity or accuracy: which suggests that the three parameters should complement, rather than substitute, one another.

Line 144-5, “we recognize there were some outliers in our data but due to the small sample size no 144 conclusive results could be determined”: This somehow contradicts the Results (line 87-89) stating that ADC of epithelioid and biphasic MPM were not significantly different. Rephrase the sentence less contradictorily.

 I am sorry to say that I did find one input error in the three biphasic MPMs. I corrected the data. Mean ADCs revealed to be 1.22 ± 0.25 × 10-3mm2/sec in MPM. The ADC (1.23 ± 0.26 × 10-3mm2/sec) of the epithelioid MPMs was not significantly higher than that (1.17 ± 0.21 × 10-3mm2/sec) of the biphasic MPMs (P = 0.73). Although Gill RR et al. [19] reported that the ADC value (1.31±0.15 × 10-3mm2/sec) of the epithelioid MPMs was statistically higher than that (1.01±0.11 × 10-3mm2/sec) of the biphasic MPMs (P=0.00024), in our study we could not repeat the results from Gill RR’s paper due to our small sample size.  

Methods, line 205: confirmed histopathologically.

 I changed to the correct form of “histopathologically.”

Line 205-6, “Pleural effusion was confirmed when cytology of the pleural effusion was negative and its culture was negative”: rephrase the sentence more clearly, right now it does not make much sense. For what was the effusion negative for? And for what was its culture negative for?

⇒  I changed these unclear sentences.

The diagnosis of MPM and pleural dissemination of lung cancers were confirmed histopathologically by biopsy or surgical procedure.  The diagnosis of empyema was confirmed by positive culture of pleural effusion or by purulent pleural effusion. The diagnosis of pleural effusion was confirmed by negative cytology and negative culture of pleural effusion.